# A Mamba-Based Foundation Model for Chemistry

## Abstract

We present a novel approach to chemical foundation models, leveraging structured state space sequence models (SSMs) to overcome the limitations of traditional Transformer-based architectures. While Transformers have achieved state-of-the-art results in chemical tasks such as property prediction and molecule generation, their self-attention mechanism is constrained by its inability to model data outside of a finite context window and its quadratic scaling with respect to window length. In contrast, SSMs offer a promising alternative for sequence modeling, enabling the capture of complex patterns and dependencies in molecular structures. Our Mamba architecture, a simplified end-to-end SSM-based neural network, eliminates the need for attention and MLP blocks, allowing for faster inference. We pre-train Mamba on a large, curated dataset of 91 million SMILES samples (equivalent to 4 billion molecular tokens) sourced from PubChem, and evaluate its performance on various benchmark datasets. Our experiments demonstrate the SSM's capacity to provide state-of-the-art results while maintaining fast inference, supporting complex tasks such as molecular property prediction, classification, molecular reconstruction, and synthesis yield prediction. This work advances the state-of-the-art in AI methodology in chemical sciences, offering a promising direction for future research in molecular modeling and discovery.

## 1 Introduction

Large-scale pre-training methodologies for chemical language models (LMs) represent a significant advancement in cheminformatics Sadybekov & Katritch (2023). These methodologies have shown impressive results in challenging molecular tasks such as predicting properties and generating molecules Ross et al. (2022). The success of these models can be attributed to their ability to learn contextualized representations of input tokens through self-supervised learning on large unlabeled corpora Bommasani et al. (2021).

Most chemical foundation models available are based on the Transformers architecture and its core attention module Pesciullesi et al. (2020); Chithrananda et al. (2020); Janakarajan et al. (2023). The efficacy of self-attention is attributed to its ability to route information densely within a context window Vaswani et al. (2017), allowing it to model complex data Tay et al. (2022). However, this property brings fundamental drawbacks as the inability to model anything outside of a finite window and quadratic scaling with respect to the window length Lin et al. (2022). A substantial amount of research has emerged on more efficient variants of attention to overcome these drawbacks Kotei & Thirunavukarasu (2023).

Structured state space sequence models (SSMs) have recently emerged as a promising class of architectures for sequence modeling Gu et al. (2021). These models can be interpreted as a combination of recurrent neural networks (RNNs) and convolutional neural networks (CNNs) Smith et al. (2022). This class of models can be computed very efficiently as either a recurrence or convolution, with linear or near-linear scaling in sequence length. Mamba is a simplified end-to-end SSM-based neural network architecture without attention or even MLP blocks Gu & Dao (2023). Mamba enjoys fast inference and linear scaling in sequence length Gu & Dao (2023).

In this study, we present a novel Mamba-based large foundation model, denoted as $O_{SMI}$-SSM-$336M$. Our $O_{SMI}$-SSM-$336M$ encoder-decoder foundation model was obtained using an efficient encoder SSM-based model aligned with an auto-encoder mechanism pre-trained on a large corpus of 91

million carefully curated molecules from PubChem Kim et al. (2023), resulting in 4 billion molecular tokens. Our main contributions are:

- We curated a dataset comprising 91M molecules from PubChem Kim et al. (2023), which is equivalent to 4B molecular tokens. We used this dataset to pre-train a large-scale Mamba-based foundation model for molecules, denoted as $O_{\text{SMI}}$-SSM-$336M$.
- We demonstrate that the inference speed of our Mamba-based model is twice the speed of a Transformer-based model in predicting HOMO-LUMO properties for 10 million samples randomly selected from PubChem while delivering state-of-the-art (SOTA) results.
- We perform extensive experimentation on several classification and regression tasks from 11 benchmark datasets, covering quantum mechanical, physical, biophysical, and physiological property prediction of small molecules.
- We evaluate the model's ability to predict chemical reaction yields in synthetic and process chemistry using the Buchwald–Hartwig cross-coupling reaction dataset. Reaction yields refer to the percentage of input materials (reactants) that are converted into output materials (products).
- We evaluate the reconstruction capacity of our $O_{\text{SMI}}$-SSM-$336M$ considering the MOSES benchmarking dataset Polykovskiy et al. (2020).

Our results section demonstrates that $O_{\text{SMI}}$-SSM-$336M$ achieves SOTA performance across various tasks, including molecular property prediction, chemical reaction yield prediction, and molecule reconstruction. Furthermore, the findings indicate that the proposed model achieve SOTA performance at higher inference speed thus offering a clear advantage over the transformer counterpart.

## 2 OVERVIEW OF THE PROPOSED APPROACH

This section provides an overview of the proposed Mamba-based $O_{\text{SMI}}$-SSM-$336M$ foundation model for chemistry. We detail the process of collecting, curating, and pre-processing the pre-training data, along with the token encoding and SMILES encoder-decoder processes. Figure 1 illustrates the general architecture of the base model.

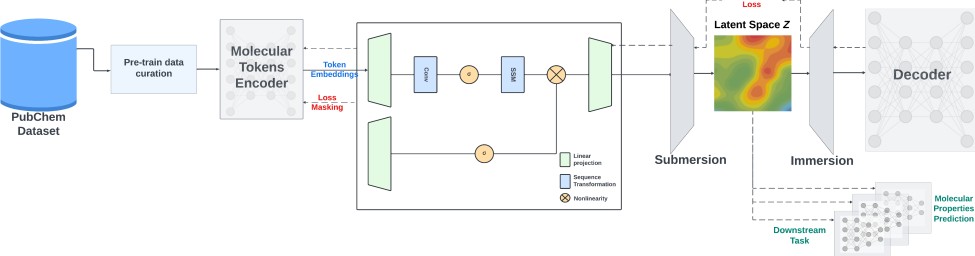

Figure 1: This figure illustrates the general architecture of the base $O_{\text{SMI}}$-SSM-$336M$ model.

### 2.1 PRE-TRAINING DATA

The pretraining data was sourced from the PubChem data repository, a public database containing information on chemical substances and their biological activities Kim et al. (2023). Initially, 113 million SMILES strings were collected from PubChem. These molecular strings underwent deduplication and canonicalization to ensure uniqueness Heid et al. (2021). Following this, a molecular transformation process was applied to validate the molecules derived from the unique SMILES strings, resulting in a final set of 91 million unique and valid molecules.

To construct the vocabulary, we utilized the molecular tokenizer proposed by Schwaller et al. (2019). The tokenization process was applied to all 91 million curated molecules from PubChem, yielding a set of 4 billion molecular tokens. From this output, we extracted 2,988 unique tokens, along with 5 special tokens. In contrast, MoLFormer, which was trained on 1 billion samples with minimal

curation, generated a vocabulary of 2,362 tokens using the same tokenization method Ross et al. (2022). This indicates that our curation process led to an enhanced vocabulary model. Detailed statistics of the pre-training dataset are provided in Table 1.

Table 1: Pre-training dataset statistics.

| Property | Mean | Std | Min | 25% | 50% | 75% | Max |
|---|---|---|---|---|---|---|---|
| Number of Atoms | 48.95 | 45.19 | 1.00 | 30.00 | 40.00 | 53.00 | 1687.00 |
| Molecular Weight (Daltons) | 344.15 | 137.79 | 1.01 | 265.32 | 330.37 | 402.47 | 18838.70 |
| LogP | 3.18 | 2.18 | -88.97 | 2.12 | 3.29 | 4.36 | 59.81 |
| Number of H-Bond Acceptors | 4.29 | 2.62 | 0 | 3.00 | 4.00 | 5.00 | 191 |
| Number of H-Bond Donors | 1.18 | 1.48 | 0 | 0.00 | 1.00 | 2.00 | 116 |
| Number of Rotatable Bonds | 4.79 | 4.09 | 0 | 3.00 | 4.00 | 6.00 | 240 |
| Topological Polar Surface Area | 67.81 | 50.11 | 0 | 40.54 | 61.77 | 84.22 | 4201.50 |
| Number of Aliphatic Rings | 0.72 | 1.07 | 0 | 0.00 | 0.00 | 1.00 | 54 |
| Number of Aromatic Rings | 1.96 | 1.24 | 0 | 1.00 | 2.00 | 3.00 | 32 |

## 2.2 MODEL ARCHITECTURE

We conduct training for $O_{SMI}$-SSM-$336M$ employing a Mamba-based encoder for tokens and an encoder-decoder architecture for SMILES embeddings space. The hyper-parameters of the model are detailed in Table 2.

Table 2: $O_{SMI}\_289M$ base architecture specificity.

| Hidden size | Layers | dt rank | d state | d conv | expand factor | dt min | dt max | dt scale | dt init floor |
|---|---|---|---|---|---|---|---|---|---|
| 768 | 24 | auto | 16 | 4 | 2 | 0.001 | 0.1 | 1.0 | 1e-4 |

| conv bias | bias | lr start | lr multiplier | Vocab size | # SMILES | # Mol tokens | # Encoder | # Decoder | Total params |
|---|---|---|---|---|---|---|---|---|---|
| True | False | 3e-5 | 1 | 2993 | 91M | 4B | 94M | 242M | 336M |

Mamba models originates from a continuous-time system that maps an input function or sequence $x(t) \in \mathbb{R}^M$ to an output response signal $y(t) \in \mathbb{R}^O$ through an implicit latent state $h(t) \in \mathbb{R}^N$ which can be mathematically formulated using the following ordinary differential equations.

$$\begin{aligned} h'(t) &= Ah(t) + Bx(t), \\ y(t) &= Ch(t) + Dx(t) \end{aligned} \tag{1}$$

where $A \in \mathbb{R}^{N \times N}$ and $C \in \mathbb{R}^{O \times N}$ control how the current state evolves over time and translates to the output, $B \in \mathbb{R}^{N \times M}$ and $D \in \mathbb{R}^{O \times M}$ depict how the input influences the state and the output, respectively.

The tokens extracted from SMILES trough the SSM encoder are embedded in a 768-dimensional space. The encoder-decoder layer is designed to process molecular token embeddings, represented as $\mathbf{x} \in \mathbb{R}^{T \times L}$, where $T$ denotes the maximum number of tokens and $L$ represents the embedding space dimension. We limited $T$ at 202 tokens, as 99.4% of molecules in the PubChem dataset contain fewer tokens than this threshold Ross et al. (2022).

In encoder-only models, a mean pooling layer is typically employed to represent tokens as SMILES in the latent space Bran & Schwaller (2023). However, this approach is limited by the lack of a natural inversion process for the mean pooling operation. To overcome this limitation, we aim to construct a latent space representation for SMILES by submersing the $\mathbf{x}$ in a latent space, denoted as $\mathbf{z}$, as described in Eq. 2.

$$\mathbf{z} = (\text{LayerNorm}(\text{GELU}(\mathbf{x}\mathbf{W}_1 + \mathbf{b}_1)))\mathbf{W}_2, \tag{2}$$

where $\mathbf{z} \in \mathbb{R}^L$, $\mathbf{W}_1 \in \mathbb{R}^L$, $\mathbf{b}_1 \in \mathbb{R}^L$, $\mathbf{W}_2 \in \mathbb{R}^{L \times L}$, with $L$ denoting the latent space size (specifically, $L = 768$). Subsequently, we can immerse $\mathbf{z}$ back by calculating Eq. 3.

$$\hat{\mathbf{x}} = (\text{LayerNorm}(\text{GELU}(\mathbf{z}\mathbf{W}_3 + \mathbf{b}_3)))\mathbf{W}_4 \tag{3}$$

where $\hat{\mathbf{x}} \in \mathbb{R}^{T \times L}$, $\mathbf{W}_3 \in \mathbb{R}^{L \times L}$, $\mathbf{b}_3 \in \mathbb{R}^L$, $\mathbf{W}_4 \in \mathbb{R}^{L \times T}$. Where $T$ representing the output feature space size (namely, $T = 202$).

A language layer (decoder) is used to process $\hat{\mathbf{x}}$, where it applies non-linearity and normalization, and projects the resulting vector into a set of logits over the vocabulary, which can then be used to predict the next token in the molecular Ferrando et al. (2023). This architecture serves as a tool for dimensionality reduction and representation learning in the domain of molecular structures.

### 2.3 PRE-TRAINING STRATEGIES

Pre-training of $O_{\text{SMI}}$-SSM-$336M$ was performed for 130 epochs through the entire curated PubChem dataset with a fixed learning rate of 3e-5 and a batch size of 128 molecules on a total of 24 NVIDIA V100 (16G) GPUs parallelized into 4 nodes using DDP and *torch run*. It involves two distinct phases: i) Learning of token embeddings through a masking process; ii) Subsequently, the token embeddings are mapped into a common latent space that encapsulates the entire SMILES string. This latent space not only facilitates the representation of the SMILES but also enables the reconstruction of both individual tokens and complete SMILES strings. Consequently, the pre-training process involves two separate loss functions: one for the token embeddings, which is based on the masking process, and another for the encoder-decoder layer, which focuses on the reconstruction of tokens. Two pre-training strategies are employed:

- In phase 1, the token encoder is initially pre-trained using 95% of the available samples, while the remaining 5% is reserved for training the encoder-decoder layer. This partitioning is necessary as the token embeddings may encounter convergence difficulties in the initial epochs, which could adversely affect the training of the encoder-decoder layer.

- In phase 2, once the token embeddings layer has achieved convergence, the pre-training process is expanded to utilize 100% of the available samples for both phases. This approach leads to an enhancement in the performance of the encoder-decoder layer, particularly in terms of token reconstruction.

For encoder pre-training we use the masked language model method defined in Devlin et al. (2019). Initially 15% of the tokens are selected for possible learning. From that selection, 80% of the tokens are randomly selected and replaced with the [MASK] token, 10% of the tokens are randomly selected to be replaced with a random token, while the remaining 10% of the tokens will be unchanged.

The adoption of different pre-training strategies has proven instrumental in enhancing the efficiency of our model, as evidenced by improvements observed in the loss functions.

### 3 EXPERIMENTS

To evaluate the effectiveness of our proposed Mamba-based model $O_{\text{SMI}}$-SSM-$336M$, we conducted experiments using a set of 11 datasets sourced from MoleculeNet Wu et al. (2018) as demonstrated in Table 3. Specifically, we evaluated 6 datasets for classification task and 5 datasets for regression tasks. To ensure an unbiased assessment, we maintained consistency with the original benchmark by adopting identical train/validation/test splits for all tasks Wu et al. (2018). We also conducted the experiments considered 10 different seeds for all the tests in other to guarantee the robustness of the approach.

We also conducted high-throughput experiments on Pd-catalyzed Buchwald–Hartwig C–N cross-coupling reactions, measuring the yields for each reaction as described in Ahneman et al. (2018). The experiments utilized three 1536-well plates, covering a matrix of 15 aryl and heteroaryl halides, four Buchwald ligands, three bases, and 23 isoxazole additives, resulting in a total of 3,955 reactions. We employed the same data splits as in Ahneman et al. (2018) to assess our model's performance with training sets of varying sizes.

To evaluate the reconstruction and decoder capabilities of $O_{\text{SMI}}$-SSM-$336M$, we utilized the MOSES benchmarking dataset Polykovskiy et al. (2020), which contains 1,936,962 molecular structures. For the experiments, we adopted the dataset split proposed by Polykovskiy et al. (2020), dividing it into training, test, and scaffold test sets, comprising approximately 1.6 million, 176,000, and 176,000 molecules, respectively. The scaffold test set includes unique Bemis-Murcko scaffolds that

Table 3: Evaluated datasets description

| Dataset | Description | # compounds | # tasks | Metric |
|---------|-------------|-------------|---------|--------|
| BBBP | Blood brain barrier penetration dataset | 2039 | 1 | ROC-AUC |
| HIV | Ability of small molecules to inhibit HIV replication | 41127 | 1 | ROC-AUC |
| BACE | Binding results for a set of inhibitors for $\beta$ − secretase 1 | 1513 | 1 | ROC-AUC |
| Clintox | Clinical trial toxicity of drugs | 1478 | 2 | ROC-AUC |
| SIDER | Drug side effect on different organ classes | 1427 | 27 | ROC-AUC |
| Tox21 | Toxicity measurements on 12 different targets | 7831 | 12 | ROC-AUC |
| QM9 | 12 quantum mechanical calculations | 133885 | 12 | Average MAE |
| QM8 | 12 excited state properties of small molecules | 21786 | 12 | Average MAE |
| ESOL | Water solubility dataset | 1128 | 1 | RMSE |
| FreeSolv | Hydration free energy of small molecules in water | 642 | 1 | RMSE |
| Lipophilicity | Octanol/water distribution coefficient of molecules | 4200 | 1 | RMSE |

are absent in the training and test sets, allowing us to assess the model's ability to generate previously unobserved scaffolds. Finally, we evaluated the inference speed of O$_{\text{SMI}}$-SSM-336$M$ by predicting HOMO-LUMO properties for 10 million samples randomly selected from PubChem.

# 4 RESULTS AND DISCUSSION

In this section, we present an analysis of the results obtained using O$_{\text{SMI}}$-SSM-336$M$ across various experiments conducted with different versions of the base model. The analysis includes: (i) A comparison between frozen and fine-tuned versions of O$_{\text{SMI}}$-SSM-336$M$, along with a comparison against state-of-the-art models on various benchmarking datasets for molecular classification and regression tasks; (ii) An evaluation of O$_{\text{SMI}}$-SSM-336$M$ for predicting chemical reaction yields; (iii) An assessment of the Decoder module using the MOSES benchmarking dataset; and (iv) A study comparing the inference speed for predicting HOMO-LUMO properties on 10 million samples randomly selected from PubChem.

## 4.1 COMPARISON WITH SOTA ON BENCHMARKING TASKS

**Results for classification tasks:** The analysis evaluates the comparative performance of O$_{\text{SMI}}$-SSM-336$M$ in its fine-tuned and frozen states relative to state-of-the-art algorithms for molecular property classification, as detailed in Table 4.

Table 4: Methods and Performance for the classification tasks of MoleculeNet benchmark datasets

| Method | Dataset | | | | | |
|--------|---------|---------|-----|------|-------|-------|
| | BBBP | ClinTox | HIV | BACE | SIDER | Tox21 |
| GraphMVP Liu et al. (2021) | 72.4±1.6 | 79.1±2.8 | 77.0±1.2 | 81.2±0.9 | 63.9±1.2 | 75.9±0.5 |
| GEM Fang et al. (2022) | 72.4±0.4 | 90.1±1.3 | 80.6±0.9 | 85.6±1.1 | **67.2±0.4** | 78.1±0.1 |
| GROVER$_{\text{Large}}$ Rong et al. (2020) | 69.5±0.1 | 76.2±3.7 | 68.2±1.1 | 81.0±1.4 | 65.4±0.1 | 73.5±0.1 |
| ChemBerta Chithrananda et al. (2020) | 64.3 | 90.6 | 62.2 | - | - | - |
| ChemBerta2 Ahmad et al. (2022) | 71.94 | 90.7 | - | 85.1 | - | - |
| Galatica 30B Taylor et al. (2022) | 59.6 | 82.2 | 75.9 | 72.7 | 61.3 | 68.5 |
| Galatica 120B Taylor et al. (2022) | 66.1 | 82.6 | 74.5 | 61.7 | 63.2 | 68.9 |
| Uni-Mol Zhou et al. (2023) | 72.9±0.6 | 91.9±1.8 | 80.8±0.3 | 85.7±0.2 | 65.9±1.3 | 79.6±0.5 |
| MolFM Zhou et al. (2023) | 72.9±0.1 | 79.7±1.6 | 78.8±1.1 | 83.9±1.1 | 64.2±0.9 | 77.2±0.7 |
| MoLFormer Chang & Ye (2024) | 73.6±0.8 | 91.2±1.4 | 80.5±1.65 | 86.3±0.6 | 65.5±0.2 | 80.46±0.2 |
| SMI-TED289M (Frozen Weights) Soares et al. (2024) | 91.46±0.47 | 93.49±0.85 | 80.51±1.34 | 85.58±0.92 | 66.01±0.88 | 81.53 ±0.45 |
| SMI-TED289M (Fine-tuned) Soares et al. (2024) | 92.26±0.57 | **94.27±1.83** | 76.85±0.89 | **88.24±0.50** | 65.68±0.45 | 81.85±1.42 |
| O$_{\text{SMI}}$-SSM-336$M$ (Frozen) | 90.81 ±0.85 | 86.36 ±0.74 | 77.04 ±0.64 | 83.83 ±0.76 | 63.52 ±0.3 | 81.42 ±0.8 |
| O$_{\text{SMI}}$-SSM-336$M$(Fine-tuned) | **92.81 ±0.27** | 90.02 ±0.5 | **83.14 ±0.34** | 86.12 ±0.96 | 63.17 ±0.75 | **83.84 ±0.2** |

Table 4 summarizes the performance of various advanced methods across several benchmarking datasets used for molecular classification tasks. O$_{\text{SMI}}$-SSM-336$M$ demonstrates comparative efficacy against Transformer-based approaches, outperforming them in three out of six datasets. Notably, O$_{\text{SMI}}$-SSM-336$M$ with its initial configuration yields results on par with current state-of-the-art methods. Further fine-tuning of O$_{\text{SMI}}$-SSM-336$M$ enhances its performance, indicating its substantial potential for accurate molecular classification and suggesting that additional performance gains may be achieved through further optimization.

**Results for regression tasks:** Subsequently, we applied O$_{\text{SMI}}$-SSM-336$M$ to the prediction of chemical properties. The performance metrics across five regression benchmarks—QM9, QM8, ESOL, FreeSolv, and Lipophilicity—are presented in Table 5.

Table 5: Methods and Performance for the regression tasks of MoleculeNet benchmark datasets. **Blue** and **Orange** indicates best and second-best performing model, respectively.

| Method | Dataset | | | | |
| --- | --- | --- | --- | --- | --- |
| | QM9 | QM8 | ESOL | FreeSolv | Lipophilicity |
| D-MPNN Yang et al. (2019) | 3.241±0.119 | 0.0143±0.0022 | 0.98±0.26 | 2.18±0.91 | 0.65±0.05 |
| N-Gram Liu et al. (2019) | 2.51±0.19 | 0.032±0.003 | 1.074±0.107 | 2.688±0.085 | 0.812±0.028 |
| PretrainGNN Hu et al. (2019) | - | - | 1.100±0.006 | 2.764±0.002 | 0.739±0.003 |
| GROVER$_{Large}$ Rong et al. (2020) | - | - | 0.895±0.017 | 2.272±0.051 | 0.823±0.010 |
| ChemBERTa-2 Ahmad et al. (2022) | - | - | 0.89 | - | 0.80 |
| SPMM Chang & Ye (2024) | - | - | 0.818±0.008 | 1.907±0.058 | 0.692±0.008 |
| MolCLR$_{GIN}$ Wang et al. (2022) | 2.357±0.118 | 0.0174±0.0013 | 1.11±0.01 | 2.20±0.20 | 0.65±0.08 |
| Hu et al. Hu et al. (2020) | 4.349±0.061 | 0.0191±0.0003 | 1.22±0.02 | 2.83±0.12 | 0.74±0.00 |
| MoLFormer Chang & Ye (2024) | **1.5894±0.0567** | 0.0102 | 0.880±0.028 | 2.342±0.052 | 0.700±0.012 |
| SMI-TED289M Soares et al. (2024) | **1.3246±0.0157** | **0.0095±0.0001** | **0.6112±0.0096** | **1.2233±0.0029** | **0.5522±0.0194** |
| O$_{SMI}$-SSM-336$M$(Frozen) | 8.9546±0.0577 | 0.0194±0.0003 | 0.8135±0.0253 | 1.6374±0.0682 | 0.746±0.0029 |
| O$_{SMI}$-SSM-336$M$ (Fine-tuned) | 2.2175±0.3194 | **0.0104±0.0001** | **0.7222±0.0139** | **1.6288±0.0347** | **0.6048±0.0023** |

Results presented in Table 5 indicate that O$_{SMI}$-SSM-336$M$ achieves performance comparable to state-of-the-art models, securing the second-best results in four of the five regression benchmarks evaluated. This demonstrates the efficacy of the Mamba-based approach in delivering results on par with Transformer-based methods, while also highlighting its robustness across a range of chemical property prediction tasks. The design of O$_{SMI}$-SSM-336$M$ aims to strike an optimal balance between predictive accuracy and inference efficiency. To exemplify this balance, we provide an analysis comparing the inference time for predicting HOMO-LUMO properties on a dataset of 10 million samples randomly selected from PubChem. This study underscores the model's capability to maintain high prediction accuracy while significantly reducing computational time, thereby offering practical advantages for large-scale chemical property predictions.

**Speed inference for HUMO-LUMO properties prediction:** To assess the inference speed of the proposed Mamba-based approach, we conducted predictions of HOMO-LUMO properties for 10 million samples randomly selected from PubChem. For comparison, we evaluated the inference time of SMI-TED289M, a Transformer-based model recognized for its state-of-the-art performance. Figure 2 illustrates the superior inference speed of O$_{SMI}$-SSM-336$M$ compared to SMI-TED289M. Specifically, SMI-TED289M required 20,606.76 seconds for HOMO property predictions and 21,038.43 seconds for LUMO property predictions using a single NVIDIA V100 32GB GPU. In contrast, O$_{SMI}$-SSM-336$M$ completed HOMO predictions in 9,735.64 seconds and LUMO predictions in 9,823.64 seconds on the same GPU. These results highlight the substantial efficiency gains of the O$_{SMI}$-SSM-336$M$ model in terms of inference speed.

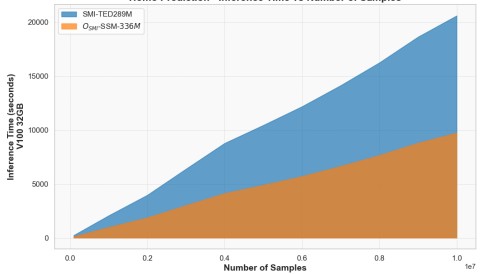 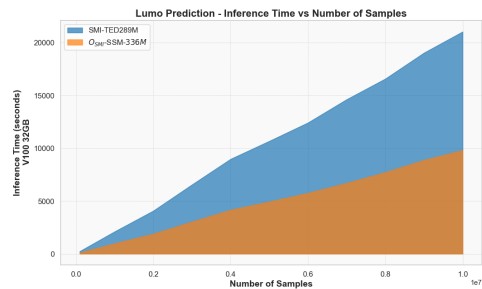

Figure 2: The figure shows the inference speed for O$_{SMI}$-SSM-336$M$ and SMI-TED289M for HOMO-LUMO predictions considering a dataset of 10M samples randomly selected from PubChem and a single NVIDIA V100 32GB GPU.

The Mamba-base approach demonstrates a substantial improvement in efficiency, being approximately 54% faster and reducing GPU usage by 6 hours, while also decreasing CO2 emissions by an average of 0.78 kg equivalent Lacoste et al. (2019). This reduction in computational resources is crucial for minimizing the environmental impact of machine learning models, which requires significant energy consumption and associated carbon footprints Rillig et al. (2023).

### 4.2 REACTION-YIELD PREDICTION

Here, we investigate the Mamba-based approach on chemical reactions. Chemical reactions in organic chemistry are described by writing the structural formula of reactants and products separated by an arrow, representing the chemical transformation by specifying how the atoms rearrange between one or several reactant molecules and one or several product molecules. Predicting outcomes of chemical reactions, such as their yield based on data gathered in high-throughput screening, is an important task in machine learning for chemistry. Fig. 3 the schema for chemical reaction.

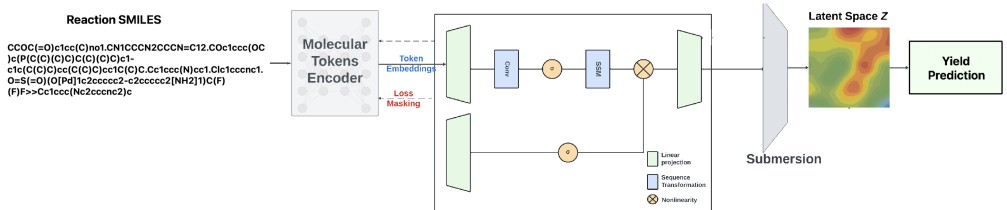

Figure 3: This figure illustrates the schema for chemical reaction yield prediction based on reaction SMILES considering the $O_{SMI}$-SSM-336$M$ model.

We assessed this architecture against state-of-the-art methods using a high-throughput dataset of Buchwald–Hartwig cross-coupling reactions, focusing on predicting reaction yields Ahneman et al. (2018). This involves estimating the percentage of reactants converted into products. Our evaluation adhered to the schema and data divisions outlined in Ahneman et al. (2018). Table 6 presents the results for the $O_{SMI}$-SSM-336$M$ model and compares its performance with existing state-of-the-art approaches.

Table 6: Performance of $O_{SMI}$-SSM-336$M$ compared with the state of the art in reaction-yield prediction on experimentally determined yields of Buchwald–Hartwig reactions through HTEs.

| Subset/Split | DFT | Yield-BERT | Yield-BERT (Aug) | DRFP | YieldGNN | MSR2-RXN | Ours |
|---|---|---|---|---|---|---|---|
| Rand 70/30 | 0.92 | 0.95±0.005 | 0.97±0.003 | 0.95±0.005 | 0.96±0.005 | 0.94±0.005 | **0.9823 ±0.0007** |
| Rand 50/50 | 0.9 | 0.92±0.01 | 0.95±0.01 | 0.93±0.01 | - | 0.93±0.01 | **0.982 ±0.0004** |
| Rand 30/70 | 0.85 | 0.88±0.01 | 0.92±0.01 | 0.89±0.01 | - | 0.90±0.01 | **0.978 ±0.0013** |
| Rand 20/80 | 0.81 | 0.86±0.01 | 0.89±0.01 | 0.87±0.01 | - | 0.87±0.01 | **0.973 ±0.0006** |
| Rand 10/90 | 0.77 | 0.79±0.02 | 0.81±0.02 | 0.81±0.01 | - | 0.80±0.02 | **0.952 ±0.0023** |
| Rand 5/95 | 0.68 | 0.61±0.04 | 0.74±0.03 | 0.73±0.02 | - | 0.69±0.03 | **0.903 ±0.0043** |
| Rand 2.5/97.5 | 0.59 | 0.45±0.05 | 0.61±0.04 | 0.62±0.04 | - | 0.57±0.05 | **0.846 ±0.0044** |
| Test 1 | 0.8 | 0.84±0.01 | 0.80±0.01 | 0.81±0.01 | - | 0.83±0.03 | **0.9827 ±0.0002** |
| Test 2 | 0.77 | 0.84±0.03 | 0.88±0.02 | 0.83±0.003 | - | 0.83±0.01 | **0.9827 ±0.0005** |
| Test 3 | 0.64 | 0.75±0.04 | 0.56±0.08 | 0.71±0.001 | - | 0.69±0.04 | **0.9823 ±0.0012** |
| Test 4 | 0.54 | 0.49±0.05 | 0.43±0.04 | 0.49±0.004 | - | 0.51±0.04 | **0.9825 ±0.0008** |
| Average 1-4 | 0.69 | 0.73 | 0.58±0.33 | 0.71±0.16 | - | 0.72±0.15 | **0.9826 ±0.0005** |

The results presented in Table 6 clearly demonstrate the superiority of the proposed Mamba-based foundation model when benchmarked against state-of-the-art methods, including gradient-boosting and fingerprint-based approaches (DRFP) Probst et al. (2022), a DFT-based random forest model (DFT) Probst et al. (2022), and transformer-based models like Yield-BERT Schwaller et al. (2021) and its augmented variant, Yield-BERT(aug.) Schwaller et al. (2021), and MSR2-RXN Boulougouri et al. (2024). The performance of the Mamba-based model can be attributed to its pre-training on an expansive dataset of 91 million curated molecules, which provides a robust foundation of chemical knowledge that significantly enhances its predictive capabilities. This pre-training enables the model to achieve high accuracy even with limited training data, as evidenced by its sustained performance when trained on just 2.5% of the available samples—a scenario where task-specific models experience a marked decline in accuracy. To ensure the robustness of our model, we conducted each experiment with 10 different random seeds.

One key observation is the model's robustness across various data splits, particularly in low-resource settings where only a small fraction of the dataset is used for training. This resilience underscores the importance of leveraging large-scale pre-training to encode generalized chemical knowledge, which can then be fine-tuned for specific tasks like reaction yield prediction. In contrast, models that are

tailored specifically for a given task tend to overfit to the nuances of the training data and struggle to generalize when the training set size is reduced, highlighting a critical limitation in their design.

Moreover, the robustness of the Mamba-based model extends to its performance on out-of-domain test sets. The ability to generalize well to data distributions that differ from the training set is a crucial aspect of model evaluation, particularly in real-world applications where the diversity of chemical reactions is vast. The Mamba-based model's consistent performance across both in-domain and out-of-domain test sets illustrates the efficacy of pre-training on a diverse and comprehensive dataset, which equips the model with the flexibility to handle a wide range of chemical environments and reaction conditions.

The comparative analysis between the Mamba-based model and other state-of-the-art methods also sheds light on the limitations of traditional approaches like DFT-based models, which, despite their theoretical grounding in quantum chemistry, may not capture the full complexity of reaction mechanisms in practical scenarios. Similarly, while transformer-based models like Yield-BERT and its augmented variant exhibit strong performance, they fall short of the Mamba-based model, particularly in low-data regimes, indicating that the sheer scale and diversity of the pre-training data play a pivotal role in achieving superior results.

These findings underscore the potential of foundation models in chemistry, where pre-training on large, diverse datasets can serve as a powerful paradigm for developing models that are not only accurate but also robust and generalizable. The implications of this work extend beyond reaction yield prediction, suggesting that similar strategies could be applied to other domains within computational chemistry and materials science, where the ability to generalize across diverse datasets is of paramount importance.

## 4.3 DECODER EVALUATION OVER MOSES BENCHMARKING DATASET

Next, conducted a comparative evaluation of the $O_{SMI}$-SSM-336$M$ model against several baseline models for SMILES reconstruction and decoding, using a test set comprising 176,000 molecules. The evaluation metrics, detailed in Table 7, provide a comprehensive view of the model's performance in key areas such as fragment similarity (Frag), scaffold similarity (Scaf), similarity to the nearest neighbor (SNN), internal diversity (IntDiv), and Fréchet ChemNet Distance (FCD).

Table 7: MOSES benchmarking dataset evaluation.

| Metric | Frag ↑ | Scaf ↑ | SNN ↑ | IntDiv ↑ | FCD ↓ |
|---|---|---|---|---|---|
| CharRNN Polykovskiy et al. (2020) | 0.9998 | 0.9242 | 0.6015 | 0.8562 | 0.0732 |
| VAE Polykovskiy et al. (2020) | 0.9984 | 0.9386 | 0.6257 | 0.8558 | 0.0990 |
| JT-VAE Jin et al. (2018) | 0.9965 | 0.8964 | 0.5477 | 0.8551 | 0.3954 |
| LIMO Eckmann et al. (2022) | 0.6989 | 0.0079 | 0.2464 | **0.9039** | 26.78 |
| MolGen-7b Fang et al. (2023) | 0.9999 | 0.6538 | 0.5138 | 0.8617 | 0.0435 |
| GP-MoLFormer Ross et al. (2024) | 0.9998 | 0.7383 | 0.5045 | 0.8655 | 0.0591 |
| $O_{SMI}$-SSM-336$M$ | **0.9999** | **0.9994** | **0.9960** | 0.8561 | **0.0025** |

The results indicate that $O_{SMI}$-SSM-336$M$ not only matches but surpasses the performance of state-of-the-art models in generating unique, valid, and novel molecules. Its near-perfect score in the Frag metric highlights its remarkable ability to retain the structural integrity of molecular fragments, a crucial aspect in ensuring the generated molecules remain chemically viable and relevant to real-world applications. This high fragment similarity, coupled with the model's low FCD score, suggests that the distribution of generated molecules closely mirrors that of natural molecules.

In addition to fragment-level accuracy, $O_{SMI}$-SSM-336$M$ demonstrates superior performance in scaffold similarity (Scaf) and nearest neighbor similarity (SNN). These metrics are particularly important in drug discovery and design, where the preservation of core molecular scaffolds is essential for maintaining biological activity. The model's ability to generate molecules with high scaffold similarity indicates that it can reliably reproduce the core structural features of molecules, which is a requirement for generating candidate compounds that retain their intended biological function.

Another significant finding is the model's performance in internal diversity (IntDiv). While high similarity scores are important, diversity within the generated set is equally crucial, especially in

scenarios where a broad exploration of chemical space is required. The $O_{SMI}$-SSM-$336M$ model achieves a commendable balance, maintaining high similarity metrics while also generating molecules with substantial pairwise dissimilarity. This capability to generate a diverse array of molecules without sacrificing structural integrity makes the model highly valuable for applications in drug discovery, where exploring a wide range of chemical possibilities is often necessary to identify optimal candidates.

Furthermore, when compared to traditional methods such as CharRNN and more advanced approaches like JT-VAE and MolGen-7b, the $O_{SMI}$-SSM-$336M$ model consistently outperforms across all evaluated metrics. This includes models like LIMO, which, despite its strong internal diversity, fails to match the other metrics, indicating a trade-off in these approaches that $O_{SMI}$-SSM-$336M$ successfully mitigates. The model's ability to achieve high scaffold similarity while maintaining diverse molecular structures suggests that its pre-training on a large-scale dataset equips it with a broad understanding of chemical space, enabling it to generalize effectively across various molecular configurations.

## 5 CONCLUSION

This paper introduces $O_{SMI}$-SSM-$336M$, a Mamba-based chemical foundation model pre-trained on a curated dataset of 91 million SMILES samples from PubChem, encompassing 4 billion molecular tokens. The model is designed to achieve high performance in evaluation metrics while decreasing the inference time.

The efficacy of $O_{SMI}$-SSM-$336M$ was rigorously assessed across a variety of tasks, including molecular property classification and prediction. The model not only achieved state-of-the-art results but also demonstrated significant efficiency improvements. Specifically, it was approximately 54% faster than existing state-of-the-art Transformer-based approaches, reducing GPU usage by 6 hours and lowering $CO2$ emissions by an average of 0.78 kg $CO2$ equivalent Lacoste et al. (2019) during the prediction of HOMO-LUMO gaps for a dataset of 10 million randomly selected samples from PubChem.

We also explored the model's capabilities in predicting chemical reaction outcomes, such as reaction yields based on high-throughput screening data, a critical task in machine learning for chemistry. The consistent performance of the Mamba-based model across both in-domain and out-of-domain test sets underscores the effectiveness of pre-training on a diverse and comprehensive dataset. This pre-training enables the model to adapt to a wide range of chemical environments and reaction conditions. Our comparative analysis revealed that while traditional approaches, such as DFT-based models, are grounded in quantum chemistry, they may not fully capture the complexity of reaction mechanisms in practical scenarios. Similarly, transformer-based models like Yield-BERT and its augmented variant, despite their strong performance, are outperformed by the Mamba-based model, particularly in low-data regimes. This highlights the critical role that large-scale, diverse pre-training data plays in achieving superior results.

Finally, we conducted a comparative evaluation of the $O_{SMI}$-SSM-$336M$ model against several baseline models for SMILES reconstruction and decoding. The model's performance across diverse metrics demonstrates the importance of leveraging large-scale dataset for pre-training, which can lead to models that not only excel in generating high-quality molecules but also possess the flexibility required to tackle complex challenges in computational chemistry and drug design.

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

# A APPENDIX

## A.1 DETAILED RESULTS - REACTION-YIELD PREDICTION

To fine-tune the $O_{SMI}$-SSM-$336M$ model, we employed a two-layer fully connected neural network. Each task was run on a single NVIDIA V100 GPU (16 GB). The fine-tuning process specifically targeted the prediction of reaction yields in Buchwald–Hartwig cross-coupling reactions, following the approach detailed in Ahneman et al. (2018), which involves estimating the percentage of reactants successfully converted into products. To ensure robustness, we repeated each experiment across 10 different random seeds, with results outlined in Table 8.

Table 8: Reaction-yield prediction results for 10 different seeds considering $SO_{SMI}$-SSM-$336M$.

| SEED | Rand 70/30 | Rand 50/50 | Rand 30/70 | Rand 20/80 | Rand 10/90 | Rand 05/95 | Rand 2.5/97.5 | Test 1 | Test 2 | Test 3 | Test 4 | Average 1-4 |
|---|---|---|---|---|---|---|---|---|---|---|---|---|
| 0 | 0.9827 | 0.9822 | 0.9791 | 0.9731 | 0.9537 | 0.9044 | 0.8406 | 0.9826 | 0.9829 | 0.9828 | 0.9828 | 0.9828 |
| 10 | 0.9805 | 0.981 | 0.9747 | 0.9742 | 0.9492 | 0.9012 | 0.8398 | 0.9831 | 0.983 | 0.9827 | 0.983 | 0.9830 |
| 20 | 0.9824 | 0.9822 | 0.9788 | 0.9731 | 0.9521 | 0.8949 | 0.8496 | 0.9829 | 0.9825 | 0.9831 | 0.9827 | 0.9828 |
| 30 | 0.9824 | 0.9819 | 0.9784 | 0.9723 | 0.9495 | 0.9066 | 0.8423 | 0.9826 | 0.9826 | 0.9826 | 0.9827 | 0.9826 |
| 40 | 0.9825 | 0.9822 | 0.9787 | 0.973 | 0.9507 | 0.9058 | 0.8438 | 0.9825 | 0.9814 | 0.9793 | 0.9826 | 0.9815 |
| 50 | 0.9827 | 0.9819 | 0.9786 | 0.9731 | 0.9537 | 0.9005 | 0.8502 | 0.983 | 0.9831 | 0.9831 | 0.9828 | 0.9830 |
| 60 | 0.9823 | 0.9818 | 0.978 | 0.9741 | 0.9553 | 0.9014 | 0.8502 | 0.9826 | 0.9827 | 0.9828 | 0.9806 | 0.9822 |
| 70 | 0.982 | 0.9821 | 0.9782 | 0.9732 | 0.9484 | 0.9111 | 0.851 | 0.9828 | 0.9828 | 0.9829 | 0.9831 | 0.9829 |
| 80 | 0.9827 | 0.9822 | 0.9787 | 0.9725 | 0.9531 | 0.9024 | 0.848 | 0.9825 | 0.9829 | 0.9814 | 0.9819 | 0.9822 |
| 90 | 0.9824 | 0.982 | 0.9785 | 0.9731 | 0.9535 | 0.9042 | 0.8421 | 0.9827 | 0.9834 | 0.9819 | 0.9832 | 0.9828 |
| Avg. | 0.9823 | 0.9820 | 0.9782 | 0.9732 | 0.9517 | 0.9033 | 0.8458 | 0.9827 | 0.9827 | 0.9823 | 0.9825 | 0.9826 |
| Std. | 0.0007 | 0.0004 | 0.0013 | 0.0006 | 0.0023 | 4E-03 | 0.0044 | 0.0002 | 0.0005 | 0.0012 | 0.0008 | 0.0005 |

## A.2 DETAILED RESULTS - SPEED INFERENCE FOR HUMO-LUMO PROPERTIES PREDICTION

Here, we present the inference speed results for predicting the HUMO-LUMO properties using 10 million samples. The comparison highlights the performance of two models: SMI-TED 289M and $O_{SMI}$-SSM-336M, focusing on their scalability as the sample size increases. In Table 9, the inference times (in seconds) for HUMO properties are reported different dataset sizes, ranging from 100k to 10M samples.

Table 9: Inference times in seconds for HUMO properties considering different dataset sizes.

| Model | 100k | 1M | 2M | 3M | 4M | 5M | 6M | 7M | 8M | 9M | 10M |
|---|---|---|---|---|---|---|---|---|---|---|---|
| SMI-TED 289M | 240.99 | 2063.72 | 3966.59 | 6389.87 | 8779.9 | 10448.99 | 12181.5 | 14136.47 | 16249.44 | 18636.85 | 20606.76 |
| $O_{SMI}$-SSM-$336M$ | 117.94 | 980.93 | 1876.18 | 3012.61 | 4126.37 | 4899.57 | 5707.83 | 6645.89 | 7657.16 | 8801.94 | 9735.64 |

Table 10, the inference times in seconds for LUMO properties are reported different dataset sizes, ranging from 100k to 10M samples. O$_{SMI}$-SSM-336M demonstrates lower inference times across all dataset sizes, making it a more efficient choice for large-scale molecular property predictions. For instance, with a dataset of 1 million samples, the inference time for O$_{SMI}$-SSM-336M (989.11 seconds) is less than half that of SMI-TED 289M (2107.62 seconds). This trend holds as the dataset size increases, with O$_{SMI}$-SSM-336M maintaining faster inference times even with 10 million samples, where it takes 9823.64 seconds compared to SMI-TED 289M's 21038.43 seconds.

Table 10: Inference times in seconds for LUMO properties considering different dataset sizes.

| Model | 100k | 1M | 2M | 3M | 4M | 5M | 6M | 7M | 8M | 9M | 10M |
|---|---|---|---|---|---|---|---|---|---|---|---|
| **SMI-TED 289M** | 246.05 | 2107.62 | 4074.55 | 6550.47 | 8979.75 | 10678.6 | 12421.24 | 14623.4 | 16578.9 | 19035.03 | 21038.43 |
| **O$_{SMI}$-SSM-336$M$** | **115.85** | **989.11** | **1895.73** | **3043.49** | **4166.82** | **4945.32** | **5762.42** | **6708.07** | **7727.71** | **8884.01** | **9823.64** |

The significant reduction in inference time offered by O$_{SMI}$-SSM-336M translates to more efficient large-scale predictions, making it a more practical choice for applications requiring the processing of millions of molecular structures. This advantage is critical in scenarios where timely predictions are necessary, such as in high-throughput virtual screening or large-scale chemical property prediction tasks. The ability to scale efficiently without sacrificing predictive performance also positions O$_{SMI}$-SSM-336M as a model better suited for deployment in computational chemistry pipelines.

