# OpenReview forum: "Mamba-based Chemical Foundational Model for Fast Inference"
_ICLR.cc/2025/Conference — ICLR 2025 Conference Withdrawn Submission_

### Official Review · Reviewer_iXmM · 2024-10-17

**Soundness:** 2
**Presentation:** 1
**Contribution:** 1
**Rating:** 3
**Confidence:** 5

**Summary:**

This paper proposes O$_\text{SMI}$-SSM, a Mamba-based foundation model for chemistry. The authors pre-train a Mamba model with 91M molecules (4B molecular tokens) based on SMILES representation. The resulting model shows reasonable performance with inference efficiency compared to Transformer-based models.

**Strengths:**

- The problem of interest, foundation model for chemistry, is an important topic in real-world applications, e.g., drug discovery.
- This paper is easy to follow.

**Weaknesses:**

- Lack of novelty.

The main contribution of this work is training a Mamba model to make a foundation model in chemistry. However, there is no new techniques in its construction. Mamba is an already proposed model, and masked pre-training objective is also popular. Therefore, this work can be viewed as applying LLM techniques to molecules, which is not enough contribution for acceptance.

---
- Imprecise motivation.

The main motivation of this work is to improve the efficiency of molecular foundation model. Although developing an efficient model is always good, there exists a trade-off between efficiency and accuracy. In general LLMs, pursuing efficiency is reasonable since they often require real-time communications. However, in chemistry, such scenarios are highly unlikely. Furthermore, accuracy is extremely important in chemical tasks since verifying the output, e.g., drug-likeness, requires expensive wet experiments. Therefore, I think "efficiency in chemical foundation models (with accuracy trade-off)" is not a reasonable direction.

---
- Lack of details.

There is no loss function description. Also, there should be detailed description on the difference between Frozen model and Fine-tuned model.

---
- About target task.

Recent chemical foundation models mainly focus on learning both language description and molecules [1,2,3]. However, this work only focuses on learning molecules which limits the chemical applications such as text-to-molecule generation.

---
[1] Edwards et al., Translation between Molecules and Natural Languages, EMNLP 2022\
[2] Pei et al., BioT5: Enriching Cross-modal Integration in Biology with Chemical Knowledge and Natural Language Associations, EMNLP 2023\
[3] Li et al., Towards 3D Molecule-Text Interpretation in Language Models, ICLR 2024

**Questions:**

1. Why did the authors only conduct experiments on 6 datasets in MoleculeNet? Most of the baselines also report the results on MUV and ToxCast.

2. What is the difference between Frozen model and Fine-tuned model?

3. How are the SMILES representations tokenized into the token space?

---

### Official Review · Reviewer_QKx8 · 2024-11-02

**Soundness:** 3
**Presentation:** 4
**Contribution:** 1
**Rating:** 3
**Confidence:** 3

**Summary:**

The work proposes to use SSMs for molecular foundation modeling. The authors pretrain on 91 million SMILES samples and evaluate performance on downstream tasks including molecular property prediction, classification, molecular reconstruction, and reaction yield prediction. The model outperforms prior transformers, GNNs/MPNNs by a large margin.

**Strengths:**

1. The work performs extensive benchmarking on multiple downstream tasks and seems to perform competitively across the board. For many tasks they outperform the baseline by a large margin.

2. They compare the efficiency of Mamba against another transformer architecture and observe large efficiency gains for large number of samples.

3. All experimental results and hyperparameter choices are clearly communicated.

**Weaknesses:**

1. The other architectures benchmarked in downstream tasks do not use the same pretraining method, so it's not exactly clear if the performance benefits are due to the Mamba architecture or due to the pre-training dataset. On the other hand, the point of this paper may be just to demonstrate a performant model.

2. It isn't a very novel idea to generate SMILES strings with sequence models. This seems to be a drop-in replacement of a transformer with Mamba.

**Questions:**

1. Can you give an explanation for how the prior algorithms were trained, what dataset they were trained on, as well as their model sizes? Otherwise it's hard to make a fair comparison.

---

### Official Review · Reviewer_CqKH · 2024-11-02

**Soundness:** 2
**Presentation:** 1
**Contribution:** 1
**Rating:** 3
**Confidence:** 4

**Summary:**

This paper investigates the application of structured state space sequence models (SSMs), specifically MAMBA, in the molecular modeling of SMILES string formats. The model is pre-trained on a large dataset of 91 million SMILES strings from PubChem, resulting in 4 billion molecular tokens, Through pretraining and finetuning, the experimental results demonstrate that the proposed methodology achieves competitive performance on prediction and generation tasks with improvements in inference efficiency.

**Strengths:**

1.   The motivation is clearly articulated. the application of Mamba to the biological domain, specifically to long-sequence data like SMILES strings, appears promising due to its ability to efficiently capture complex patterns and dependencies within these sequences.The authors effectively establish the need for an alternative to Transformer-based architectures in handling long-sequence data in chemistry.

2.   Experimental results on tasks such as property prediction, reaction yield prediction, and molecular generation show positive outcomes.

**Weaknesses:**

1.   The contribution is overlapped with the work from [1], lots of accliam and desription are same from the work. which make the contribution of this work is marginal and unclear.
2.   There is a disconnect between the stated motivation(deal with long-sequence) and the experimental design. Specifically, the experiments are primarily conducted on molecules with lengths of 49 ± 45, which is considerably shorter than the maximum sequence length supported by transformer-based models (e.g., 512 for ChemBERTa [1]). Although inference speed improvements are reported, experiments on longer sequences, such as proteins, would provide stronger evidence for the model’s applicability to long-sequence tasks.
3.   The manuscript lacks sufficient details regarding the experimental implementation. Several key aspects require clarification (please see questions below).

[1] Eduardo Soares, Victor Shirasuna, Emilio Vital Brazil, Renato Cerqueira, Dmitry Zubarev, and
Kristin Schmidt. A large encoder-decoder family of foundation models for chemical language.
arXiv preprint arXiv:2407.20267, 2024.
[2] Seyone Chithrananda, Gabriel Grand, and Bharath Ramsundar. Chemberta: large-scale selfsupervised pretraining for molecular property prediction. arXiv preprint arXiv:2010.09885, 2020.

**Questions:**

* "In Section 2.2, the authors propose using a language decoder alongside MAMBA's inherent decoding capabilities. This design choice appears to contradict the paper's efficiency claims, as self-attention mechanisms typically incur higher computational costs compared to MAMBA modules. Could the authors justify this architectural decision?"

* "Figure 1's architectural representation requires clarification. The diagram suggests the SSM module is integrated as a standalone component following the molecular encoder's embeddings. The authors should specify whether the term 'MAMBA-based encoder' (line 124) indicates MAMBA modules are:
a) supplementary components to an existing encoder (e.g., transformers, GNNs), or
b) fundamental building blocks comprising the encoder's internal layers."

* "Section 4.3 would benefit from a detailed description of the generation task implementation, specifically addressing the input format and the utilization of pretrained model components."

* "While MAMBA is renowned for circumventing the cubic complexity associated with sequence length in self-attention mechanisms, the incorporation of a language decoder during pretraining (Section 4.3) raises questions about computational efficiency. How do the authors reconcile these seemingly contradictory design choices?"

* "Regarding Section 4.2, the superior generalization performance of MAMBA-based representations warrants further analysis. The results suggest this advantage stems from the MAMBA architecture itself rather than large-scale pretraining, given that transformer-based pretrained baselines demonstrate comparatively lower generalization capabilities."

* "The authors' description in Section 2.3 of the two-phase strategy, particularly regarding dataset partitioning, is described as counter-intuitive. This warrants a more comprehensive analysis and justification of the approach."

---

### Official Review · Reviewer_99Vu · 2024-11-04

**Soundness:** 2
**Presentation:** 2
**Contribution:** 2
**Rating:** 5
**Confidence:** 4

**Summary:**

This paper proposes a MAMBA-based model for regression and classification tasks in chemistry. This model called O_{SMI}-SSM-336M is an encoder-decoder model operating on SMILES strings. It is first pre-trained in a two stage process: (i) initially using a BERT-like loss on the masked input tokens, before (ii) a reconstruction loss is used with the decoder. The resultant model's encoder's weights are then fine-tuned on particular regression/classification tasks. The authors show how the method achieves excellent regression/classification performance, while operating at a faster inference speed than a comparable transformer-based model,

**Strengths:**

# Strong Empirical Results
O_{SMI}-SSM-336M obtains strong empirical results on a range of regression and classification tasks. It often performs the best or second best out of the models considered. While it would have been nice to have seen the tradeoffs this incurs (e.g., parameter counts compared to baselines), this strong predictive performance suggests that this model works well.

# Compelling low-data results
In the reaction yield experiment (Table 6), the model performs much better than the baselines in the low data regime, an important and common problem.

**Weaknesses:**

# Modeling choices not investigated/ablated
An ablation into frozen weights and fine-tuning is included as part of the main results (Tables 4 and 5); however, the effects of the two stages of pre-training are not empirically evaluated. How was the procedure detailed on lines 190-193 derived and was this assessed empiracally?

# Details and experimental setup is hard to follow
I felt the paper would have benefited from further details of the model and approach. The Mamba model is briefly described in Section 2.2, but the explanation is very high-level. One of the advantages of Mamba is its linear scaling with large sequence length, but this does not seem to actually be necessary here as the SMILES sequences are generally quite small? Therefore, I was confused about the motivation behind the model.

I also found the experiments difficult to understand. Some points:
* What is the metric used in Table 6? (It would be helpful to include the experimental setup at least in the appendix rather than deferring to another paper).
* In Table 4, MolFormer is reported as obtaining a score of 73.6 on the BBBP task, which differs from its score in the original paper  (Table 1, Ross et al., 2022). (Also the citation is incorrect). Is the experimental setup different?
* The speedup shown in Figure 2 is interesting, but hard to judge when presented without the predictive performance results. Is the performance between the two models similar?
* Section 4.6 describes how O_{SMI}-SSM-336M generates many unique and novel molecules (line 419). However, my understanding of this task is that you are trying to reconstruct "known" molecules from MOSES, so isn't generating unique and novel molecules instead disadvantageous?

# Novelty is low
I thought the paper's novelty was low and similar to previous approaches such as MolFormer (Ross et al., 2022), which also used a BERT-like loss to pretrain a model for subsequent use on regression and classification tasks. The switch to a Mamba architecture and two-stage pre-training regime seems fairly straightforward.

**Questions:**

Please see my main questions in the weaknesses section above. Other questions:
1. Is the decoder also a Mamba-based architecture or do you just use a Transformer-based model for this part?
2. What is the bottom linear projection needed for in Figure 1?
3. Table 2's caption suggests that 289M parameters were used, but elsewhere in the text the figure 336M is used instead. How many parameters are there in total and were any experiments done on different sized models?

**Details Of Ethics Concerns:**

This paper has high overlap with another paper I am reviewing for this conference. Even though the models differ (slightly), large amounts of text in the two papers are identical. (I have made a separate comment to the AC so that they can follow up on this). Given the high overlap between the two papers, my two reviews are also very similar.

---

### Note · Authors · 2024-11-25

I have read and agree with the venue's withdrawal policy on behalf of myself and my co-authors.